# Adult Pleomorphic Rhabdomyosarcoma: Case Report

**DOI:** 10.3390/reports8030166

**Published:** 2025-09-01

**Authors:** Beatrice Oancea, Roxana Elena Mirică

**Affiliations:** 1Social Insurance Medicine Office, 020021 Bucharest, Romania; cirsteaabeatrice11@gmail.com; 2Faculty of Medicine, ”Carol Davila” University of Medicine and Pharmacy, 020021 Bucharest, Romania; 3Regina Maria, Private Healthcare Network, 077190 Bucharest, Romania

**Keywords:** pleomorphic rhabdomyosarcoma, surgery, metastasis, chemotherapy, palliative radiotherapy

## Abstract

**Background and Clinical Significance:** Rhabdomyosarcoma (RMS) is a rare and aggressive malignant soft-tissue sarcoma (STS) arising from skeletal connective tissues and is most commonly seen in the pediatric population. The pleomorphic subtype is mostly seen in adults in the sixth and seventh decades of life, representing 1% of all histological types of RMS and having a very poor prognosis. **Case Presentation:** This report presents the case of a 63-year-old male with a medical history of papillary thyroid cancer, who presented with an ulcer-hemorrhagic malignant tumor, namely, a poorly differentiated desmin-positive pleomorphic rhabdomyosarcoma (PRMS), with impressive dimensions located on the posterior thoracic wall. This tumor was surgically removed via a wide resection, followed by palliative chemotherapy and radiotherapy. However, the patient relapsed locally, with pulmonary, bone, and lymph node metastases. The peculiarity of this case is represented by the rapid growth, aggressive nature, and high metastatic potential of the adult RMS, as well as its poor response to treatment. **Conclusions:** The presented case underscores the need for early diagnosis, multidisciplinary management, and exploration of molecular profiling for therapeutic planning.

## 1. Introduction and Clinical Significance

The incidence of rhabdomyosarcoma (RMS) has been estimated at about 3% of all adult soft-tissue sarcomas [1]. Pleomorphic rhabdomyosarcoma (PRMS), also known as anaplastic rhabdomyosarcoma, represents the most common subtype of RMS. It usually affects adults, while being very rare in children [2]. According to the WHO classification, RMS can be subdivided into four major histologic variants: embryonal, alveolar, pleomorphic, and spindle cell/sclerosing RMS. Pleomorphic rhabdomyosarcoma (PRMS) differs significantly from the embryonal and alveolar subtypes, which are more common in children. Unlike pediatric rhabdomyosarcomas, PRMS presents predominantly in older adults and is characterized by greater genomic complexity, poor response to standard chemoradiotherapy, and an aggressive clinical course. Its diagnosis relies heavily on histological examination and immunohistochemical confirmation due to its undifferentiated morphology. Therapeutically, PRMS behaves more like other high-grade adult soft-tissue sarcomas and lacks standardized treatment protocols, often requiring a combination of surgery, radiotherapy, and palliative chemotherapy with limited efficacy. The pleomorphic type has worse prognosis when compared with other sarcomas [3]. The American College of Surgeons has presented the anatomic distribution of soft-tissue sarcomas in adults as follows: almost 50% occur on the thigh and buttock, followed by the torso, upper extremities, retroperitoneum, and head and neck [4]. Major clinical nomograms of survival rate allowing for estimation of the prognosis of patients with soft-tissue sarcomas include stage, histological grade (an independent indicator of the degree of malignancy), tumor size (which is directly proportional to the risk of development as well as local and distant recurrence), age, anatomic site, and histologic subtype [5,6]. Unlike pediatric RMS, which has seen improvements in survival due to standardized treatment protocols, adult RMS continues to have a poor prognosis due to delayed diagnosis and limited response to conventional therapies [7]. The National Cancer Institute indicated a worse five-year survival rate [8]. The case presented in this study stands out due to its unusually rapid post-surgical progression, significant tumor volume, and extensive metastatic spread within a short timeframe. It exemplifies the aggressive nature of adult PRMS and underscores the diagnostic and therapeutic challenges it poses, especially in resource-limited or late-presentation settings.

## 2. Case Presentation

We present the case of a 63-year-old male with a known history of papillary thyroid carcinoma, who was treated in 2013 with total thyroidectomy pT3mpNx, R1, followed by radioactive iodine therapy, and was considered to be in remission. The patient presented in June 2018 with a rapidly enlarging, ulcerated tumor on the posterior thoracic wall. Objective examination revealed a conscious, cooperative, and temporally and spatially oriented patient with ECOG 2 performance status, chronic respiratory failure, SaO_2_ at 95%, and blood pressure at 130/80 mmHg. The local examination showed the formation of a PRMS of impressive dimensions located in the posterior region of the thorax, with the absence of the thoracic wall on ½ of the tumor size, showing vegetative, ulcerated, and hemorrhagic lesions (Figure 1). As such, the patient underwent surgical resection. No preoperative needle or incisional biopsy was performed due to the urgent presentation and hemorrhagic character of the lesion. As such, comprehensive preoperative staging with full-body imaging was not performed prior to surgery. Staging was completed postoperatively, when distant metastases (lungs, lymph nodes) were identified.

Intraoperatively, the tumor was found to infiltrate the muscular layers of the thoracic wall, including portions of the latissimus dorsi and paraspinal muscles. Despite the involvement of adjacent soft tissues and encasement of spinous processes, the vertebral bodies remained structurally intact. The patient retained sufficient spinal stability and did not require an external supportive corset postoperatively.

A wide resection was performed, which resulted in a large soft tissue defect over the posterior thoracic wall. Reconstruction was performed using a rotational latissimus dorsi musculocutaneous flap, which ensured adequate coverage. Postoperative healing was uneventful, with no major flap-related complications.

The diagnosis of PRMS was established postoperatively based on histological and immunohistochemical findings, which revealed positive staining for desmin and MyoD1 (myogenic differentiation 1) and negative staining for h-caldesmon and SMA (smooth muscle actin), helping to exclude smooth muscle tumors such as leiomyosarcoma. Desmin is a muscle-specific protein (i.e., a type III intermediate filament) [9], while myogenin is a transcription factor which induces myogenesis and myogenic differentiation 1 is a protein that plays a major role in muscle development and repair [10,11]. The number of mitoses was 4–5 at 10× magnification and the Ki-67 proliferation index was 40% (July 2018), consistent with a high-grade tumor (Figure 2). Nuclear staining for myogenin was also observed, further supporting the diagnosis of PRMS.

One month after surgery, the patient returned to the clinic due to right axillary pain, with palpation of a 5 cm adenopathic block. An imaging workup included evaluation of the primary site as well as sites of potential metastatic spread (CT scan of trunk, head, and neck), which revealed local tumor relapse, regional lymph node involvement, and lung metastasis. It was decided to follow up with palliative chemotherapy (Gemcitabine 900 mg/mp iv + Docetaxel 35 mg/mp iv) for three months.

Four months after surgery and three months after the initiation of palliative chemotherapy, the patient visited our department for a radiotherapy session.

In October 2018, a thoracic abdominal–pelvic CT scan was performed with contrast material. The imaging aspect was aggravated from the previous examination (July 2018) by the significant dimensional progression of the mass, with the malignant tumor’s CT aspect centered at the level of the muscular soft parts corresponding to the posterior thoracic lumbar region (level T7-D2), with invasion of the tegument, subcutaneous fat, and embedding of the spinal processes of the T9–T11 vertebral bodies, axial diameters (197/46 mm), and cranio-caudal diameter (212 mm–11/46 anterior). The presence of central necrotic areas at the level of the tumor mass was also described (Figure 3).

The significant progression of suggestive lesions for secondary determinations with diffuse distribution in the pulmonary parenchyma, pulmonary pleura, and mediastinal pleura were reported, along with the occurrence of pleural fluid in a small quantity bilaterally (Figure 4).

The significant dimensional progression of the right axillary tumor adenomegaly, associated with extensive central necrosis, was also determined (Figure 5), along with an osteocondensation-focused lesion centered on the left pubic branch with a suspicious aspect for secondary determination.

As a case of PRMS exceeding the therapeutic resources for curative purposes, the patient was recommended for palliative radiotherapy. Palliative electron teleradiotherapy was initiated for hemostatic purposes on the target volume: the posterior thoracic tumor (30 Gy, 3 gy/fr). The patient underwent four fractions, up to a total dose DT = 12 Gy, with mediocre tolerance and without clinical response. During this period, the patient’s condition continued to deteriorate gradually, requiring continuous emergency hospitalization for chronic acute respiratory failure (SaO_2_ at 89%), marked dyspnea, high blood pressure (140/100 mmHg), important fatigue, abundant sweating, generalized edema, and acute psychotic episodes with auditory hallucinations. As such, the patient received symptomatic, supportive, and local hemostatic treatment.

A chest X-ray revealed bilateral diffuse projected alveolite opacities and possibly bilateral mixed pulmonary infiltrates. An interstitial drawing of reticulo-micronodular type accentuated the bilaterally diffuse opacities and small pleural fluid overflow. The medical team continued to offer the patient supportive care, but he deceased four months after the diagnosis. The patient’s prognosis was very poor, with the following evidence: tumor size T4, regional lymph node metastasis N1, distant metastasis M1, unfavorable site (trunk), and histologic grade 3 (poor differentiated, high grade), all leading to the stage IV disease.

The differential diagnosis of a soft-tissue sarcoma includes both benign and malign tumors, such as lipomas, epidermoid cysts, schwannomas, lymphomas, and metastatic disease [13]. Malignant fibrous histiocytoma (MFH) (sometimes considered “pleomorphic sarcoma, not otherwise specified”) and pleomorphic leiomyosarcoma (LMS) may also be considered in the differential diagnosis. MFH, although known to occasionally express both desmin and SMA, should not express other specific skeletal muscle markers such as MyoD1, fast skeletal muscle myosin, myf4 (myogenin), and myoglobin. LMS, a myoid tumor with desmin expression, morphologically has intersecting fascicles, lacks the presence of large polygonal rhabdomyoblasts, and does not express specific skeletal muscle markers [14].

## 3. Discussion

PRMS is currently defined as a high-grade sarcoma, an aggressive lesion arising in the deep soft tissues of the extremities with a high propensity for metastasis [15]. For the great majority of soft-tissue sarcomas there is no known etiology, but several associated or predisposing factors can be attributed to the following environmental means:Radiation exposure (PRMS being the most frequent histological type of radiation-induced soft-tissue sarcoma);Chemical exposure (vinyl chloride, arsenic, phenoxy herbicides, chlorinated dioxins) [16];Viruses (human herpesvirus 8 and Epstein–Barr) [17];Genetic factors—certain clinical syndromes are associated with a genetic predisposition for the development of sarcoma (e.g., Li–Fraumeni syndrome, Werner syndrome, neurofibromatosis type 1, Garner syndrome, Becwith–Wiedemann syndrome, Costello syndrome) [17].

PRMS is one of the main types of soft-tissue sarcoma with nonspecific genetic alterations. Frequently mutated genes include the TP53 tumor suppressor, type 1 NF1, and alpha-thalassemia/mental retardation syndrome X-linked (ATRX); the latter one has been correlated with the alternative lengthening of telomeres [17,18]. Recent genomic profiling efforts have begun to elucidate the molecular landscape of adult RMS. In a recent study, uncommon adult RMS entities, including PRMS, were found to harbor distinct genetic alterations, opening the door to targeted therapies [19]. Additionally, a novel RAB3IP–HMGA2 fusion transcript has been identified in adult rhabdomyosarcoma of the head and neck, suggesting potential diagnostic and therapeutic implications as RAB3IP is associated with cellular proliferation, and HMGA2 with increased metastasis rate and poor prognosis [20]. Another study identified the up-regulation of MMP13 and WNT7B (which influence oncogenesis) in undifferentiated pleomorphic sarcomas, their presence suggesting the poor differentiation of this subtype. Recently, molecular techniques have highlighted a translocation involving chromosomes t(4; 19) (q35; q13) or t(10, 19)(q26; q13) [21].

Regarding the histology, PRMS is characterized by pleomorphic rhabdomyoblasts: multinucleated giant cells with hyperchromatic nuclei and atypical mitotic figures [22], where their nuclear size three-fold larger than that in “typical” tumor cells. Positive nuclear staining for desmin, muscle-specific actin, and myogenin (Myf4) on immunohistochemistry are found in over 95% of such tumors [23].

The American College of Surgeons Patterns of Care Study for adult STS showed that 23% of patients had metastatic disease at presentation; PRMS usually presents as a large deep-seated mass with early metastasis, especially to the lungs [], and a complete pulmonary metastasectomy was feasible in ⅓ of the patients [24]. The United Kingdom Department of Health has published criteria for patients with soft-tissue lesions: a tumor >5 cm; a painful lump that is increasing in size and deep to the muscle fascia; and recurrence after previous excision [25]. Our patient met all these criteria.

The College of American Pathologists, the 2017 American Joint Committee on Cancer (AJCC)/Union for International Cancer Control (UICC) cancer staging manual, and the French Federation of Cancer Centers Sarcoma Group (FNCLCC) incorporate differentiation, mitotic rate, and extent of necrosis for the determination of grade 1 (well differentiated, low grade), grade 2 (moderately differentiated), or grade 3 (poorly differentiated, high grade) [26].

The diagnosis of a suspicious soft-tissue mass involves cross-sectional imaging, such as MRI for the evaluation of the extremities, trunk, head, and neck; a CT for retroperitoneal and visceral sarcomas; and a PET scan for prognostication, grading, and determining the response to neoadjuvant chemotherapy [26]. A recent study has proposed a multiagent therapy with Vincristine and Irinotecan for patients with metastases, which also includes cycles with reduced doses of doxorubicin and cyclophosphamide alternatively in association with Ifosfamide and etoposide [27].

Histological examination of the lesion via core needle biopsy is essential for diagnosis. As the diagnosis of RMS is often difficult to formulate, unplanned and inappropriate surgical interventions are frequently performed in medical clinics before a definitive diagnosis is established [28]; this was also the case for our patient.

The treatment of RMS includes chemotherapy for primary cytoreduction and eradication of metastatic disease, radiotherapy for local residual disease, and surgical resection [29]. Chemotherapy regimens for high-grade soft-tissue sarcoma include Doxorubicin (25 mg/m^2^/day), Ifosfamide (2000–3000 mg/m^2^/day), and Mesna (1200–1800 mg/m^2^/day) for 21 days [29]. Radiotherapy is recommended for all patients with RMS; according to relevant guidelines, the preoperative dose for metastatic disease is 50.4 Gy and the postoperative radiation dose is between 60 and 75 Gy for all sites [29,30].

In almost all cases, appropriate surgical resection is a prerequisite for the curative treatment of soft-tissue sarcoma, with varying levels of success. These procedures include marginal resection or excisional biopsy, wide resection (an intermediate procedure), and radical resection. Chest wall tumors tend to be associated with poor results after surgical excision if the margins are positive [29]. Neoadjuvant chemotherapy is individualized in patients with more common adult-type RMS. In those treated with initial surgery and with higher-risk tumors (≥5 cm, grade 2 or 3, tumors located deep to fascia, locally recurrent tumors, positive margins at surgery), undergoing adjuvant chemotherapy (taking into consideration the patient’s performance status), and certain comorbid factors (including age, tumor size and location, histological subtype, and toxicity) [31], the prognosis is unfavorable. In our case, the patient presented all these risk factors, limiting the administration of higher chemotherapeutic doses and stronger combinations. Recent studies have explored the potential role of immunotherapy (pembrolizumab) in advanced soft-tissue sarcomas, including PD-1/PD-L1 checkpoint inhibitors. Although rhabdomyosarcomas generally exhibit a low mutational burden and limited immunogenicity, small studies and case reports have suggested that specific molecular subtypes may benefit from immunotherapy [32]. Ongoing clinical trials aim to better define these subsets and the predictive biomarkers involved. Although the advanced stage and poor performance status of the patient precluded the use of immunotherapeutic agents in the present case, their consideration in future protocols is warranted.

In our case, the rapid progression and poor performance status limited the scope of treatment to palliative intent. In patients diagnosed earlier or with better ECOG scores, neoadjuvant chemotherapy, radical surgical resection with negative margins, and adjuvant radiotherapy may improve outcomes. Molecular profiling to identify actionable mutations or fusion genes could also open new therapeutic options. Unfortunately, such interventions were not feasible in this case due to the fulminant course and extent of disease at presentation.

Many reports have evaluated patient and tumor characteristics to determine prognostic factors for disease-free survival (DFS), overall survival (OS), and local recurrence (LR). The most powerful predictor of DFS and OS is the AJCC TNMG stage of the tumor, with stage 4 disease reflecting poor prognostic implications [26]. The average overall survival in adults is 18 months; in our case, the patient died approximately 6 months after diagnosis.

The overall five-year survival is about 30%, that with localized disease is 35%, and that with metastatic disease is 11% [33].

The presented case highlights the diagnostic and therapeutic challenges associated with adult PRMS in a real-world context. The aggressive progression, rapid relapse, and poor responsiveness to standard therapies emphasize the need for early recognition and multidisciplinary coordination. While PRMS is rare, its clinical behavior can be devastating, as illustrated in this patient’s outcome. Our report adds to the growing body of evidence that PRMS requires individualized management strategies and underscores the urgent need for research into targeted and immunotherapeutic options. Recognizing such patterns may lead to earlier interventions and potentially better outcomes in similarly aggressive cases.

### Limitations and Future Directions

The main limitation of this case report is that it describes a single patient, which inherently restricts the generalizability of its findings. Furthermore, the absence of preoperative biopsy and comprehensive imaging prior to surgery limited the ability to establish an accurate initial staging and to plan an optimal therapeutic approach. Another limitation is the lack of access to molecular profiling and immunotherapy, which may have provided additional insights into potential targeted therapeutic options. Current evidence on pleomorphic rhabdomyosarcoma in adults is scarce, based largely on retrospective series and heterogeneous treatment protocols, which hampers the development of standardized management strategies.

Future studies should focus on multicenter collaborations and prospective registries to better characterize the clinical behavior, prognostic factors, and treatment outcomes in adult PRMS. Advances in molecular profiling and the integration of novel therapies such as targeted agents and immunotherapy should also be further explored. Early recognition, multidisciplinary evaluation, and timely integration of palliative care remain essential, particularly given the aggressive course and poor prognosis of PRMS in adults.

## 4. Conclusions

We presented a case of PRMS that started in the sixth decade of the patient’s life, with rapid post-surgical recurrence and metastasis to the lungs, lymph nodes, and bone; all these characteristics are specific, according to the literature. The peculiarity of this case is represented by the tumor’s dimensions, appearance, and the fulminant evolution toward exitus.

Due to the rarity of soft-tissue sarcomas and the numerous forms in which they can present with respect to tumor histology, site, and size, there are many options for their optimal management. Therefore, the delivery of treatment requires a multidisciplinary team that includes experienced pathologists, radiologists, surgeons, and oncologists. The next generation of molecular genetic testing is expected to serve as an important tool for determining appropriate treatment modalities, leading to improved prognosis for patients with all types of soft-tissue sarcomas.

## Figures and Tables

**Figure 1 reports-08-00166-f001:**
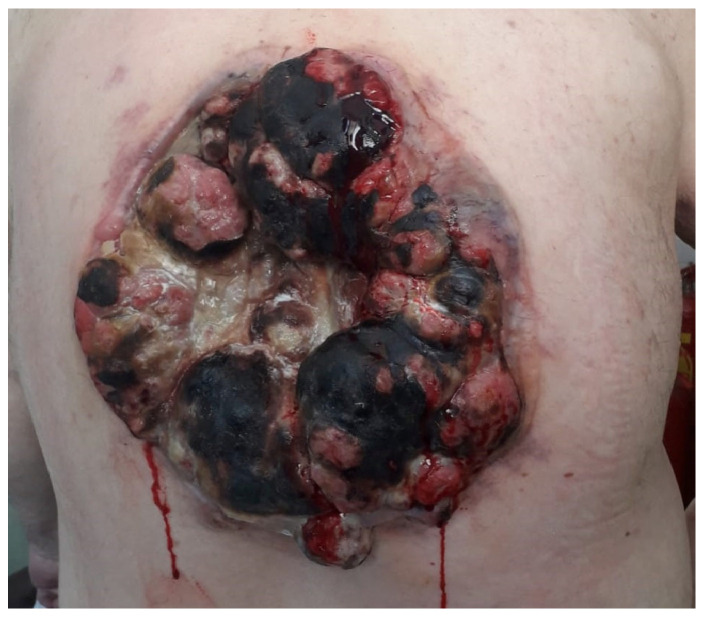
Tumor mass located at the posterior thoracic wall, with a cauliflower-like appearance and with areas of necrosis and bleeding.

**Figure 2 reports-08-00166-f002:**
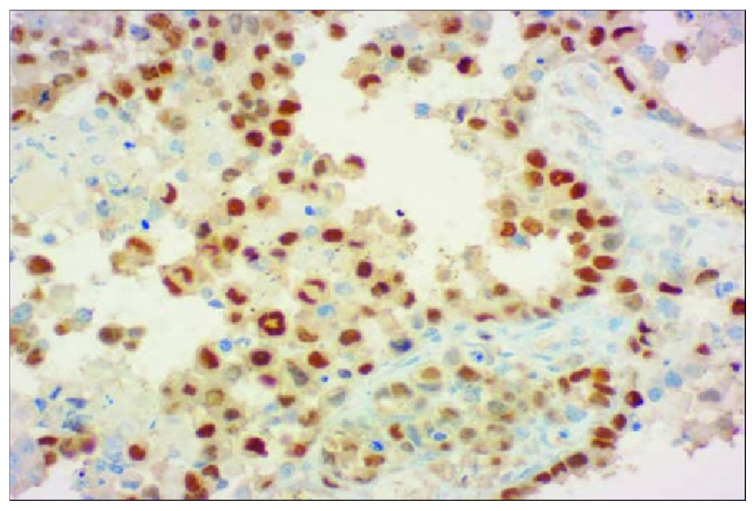
Positive immunohistochemical staining for desmin, IHC, ×40 magnification [12] Reprinted from Ref. [12].

**Figure 3 reports-08-00166-f003:**
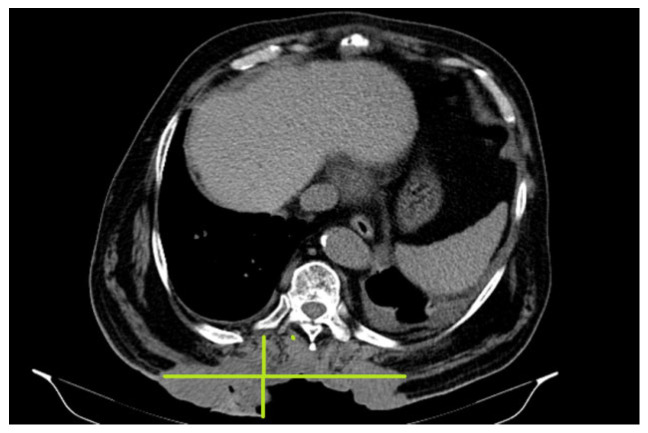
Malignant tumor CT aspect centered at the level of the muscular soft parts corresponding to the posterior thoracic lumbar region (level T7-D2), currently with invasion of the tegument, subcutaneous fat and embedding of the spinal processes.

**Figure 4 reports-08-00166-f004:**
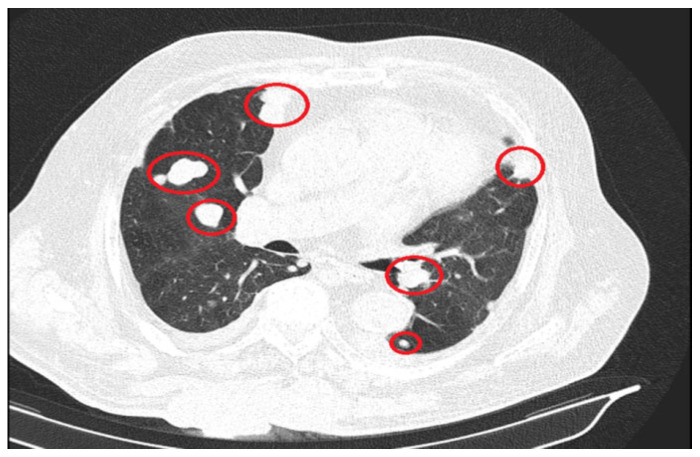
Secondary determinations located in both lung parenchyma.

**Figure 5 reports-08-00166-f005:**
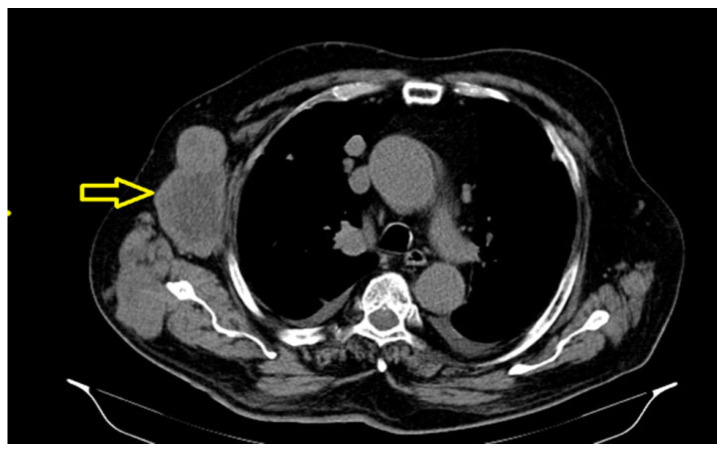
The important dimensional progression of the right axillary tumor adenomegaly, currently associated with extensive central necrosis.

## Data Availability

The original contributions presented in this study are included in the article. Further inquiries can be directed to the corresponding author.

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
