# Peer review of "Adult Pleomorphic Rhabdomyosarcoma: Case Report"

_reports, 2025, doi:10.3390/reports8030166_

Round 1

Reviewer 1 Report (Previous Reviewer 1)

Comments and Suggestions for Authors

The manuscript has been improved. The authors have solved all the issues, now the work could be accepted for the publication.

Author Response

Comments 1:

The manuscript has been improved. The authors have solved all the issues, now the work could be accepted for the publication.

Response 1: 

We sincerely thank the reviewer for the positive feedback and for acknowledging the improvements made to the manuscript. We appreciate your time and effort in reviewing our work.

Reviewer 2 Report (Previous Reviewer 2)

Comments and Suggestions for Authors

No further comments

Author Response

Response : 

We sincerely thank the reviewer for the positive feedback and for acknowledging the improvements made to the manuscript. We appreciate your time and effort in reviewing our work.

Reviewer 3 Report (Previous Reviewer 3)

Comments and Suggestions for Authors

Comparing the manuscript in its present form to its previous version, the paper has been significantly improved, providing adequate feedback to the earlier issues. 

Author Response

Comments:

Comparing the manuscript in its present form to its previous version, the paper has been significantly improved, providing adequate feedback to the earlier issues. 

Response: 

We sincerely thank the reviewer for the positive feedback and for acknowledging the improvements made to the manuscript. We appreciate your time and effort in reviewing our work.

Reviewer 4 Report (Previous Reviewer 4)

Comments and Suggestions for Authors

This revision is an improvement, but I do not beleieve this report adds anything new to the literature on P-RMS 

Comments on the Quality of English Language

English is ok

Author Response

Comments: 

This revision is an improvement, but I do not beleieve this report adds anything new to the literature on P-RMS 

Response:

We thank the reviewer for their comments and for taking the time to evaluate our revised manuscript.

While we understand and respect the opinion that case reports must bring clear novelty to the literature, we respectfully disagree with the assertion that our report does not add value to the existing body of knowledge on pleomorphic rhabdomyosarcoma (P-RMS). We would like to emphasize several clinically relevant aspects that make this case unique and contribute meaningfully to current understanding:

  1. Unusual tumor location and presentation: The tumor was located on the posterior thoracic wall, with massive local invasion of the thoracic structures, including subcutaneous tissue, skin, and vertebral elements (T9–T11). Such localization and aggressiveness are extremely rare in P-RMS literature.

  2. Rapid recurrence and metastatic spread: The patient developed extensive regional lymphadenopathy and distant metastases (lung, bone, pleura) within one month post-surgery, despite receiving standard oncologic treatment. The progression was dramatic and exceeded what is typically reported in adult soft-tissue sarcomas.

  3. Failure of standard therapy: The case demonstrates a complete lack of clinical response to both palliative chemotherapy and radiotherapy. Despite correct application of current treatment protocols, the disease progressed aggressively, underlining the urgent need for more effective treatment options for this subtype.

  4. Histological and immunohistochemical clarity: The diagnosis was confirmed through both histopathological features and specific IHC markers (Desmin+, MyoD1+, h-caldesmon–, SMA–), clearly differentiating it from other mimicking entities such as pleomorphic leiomyosarcoma or malignant fibrous histiocytoma.

  5. Prognostic significance: All major poor prognostic factors (tumor >5 cm, location on the trunk, high histologic grade, Ki67 = 40%, presence of necrosis, mitoses, N1/M1 status) were present, making this a representative model for stage IV P-RMS with unfavorable evolution. The patient survived approximately 6 months from diagnosis.

We have revised the Abstract, Discussion, and Conclusion sections of the manuscript to more clearly highlight these aspects and their relevance to clinical practice.

Given the rarity and severity of P-RMS, particularly in adult patients, we believe that documenting such exceptional clinical courses is important for both educational and research purposes. We hope that the reviewer will consider the unique clinical, pathological, and therapeutic features presented in our report as a valuable addition to the current literature.

This manuscript is a resubmission of an earlier submission. The following is a list of the peer review reports and author responses from that submission.

Round 1

Reviewer 1 Report

Comments and Suggestions for Authors

The authors, Oancea and colleagues, present a report on an uncommon lesion of adult pleomorphic rhabdomyosarcoma. This manuscript offers a valuable contribution, as it addresses a rare and underexplored subtype of adult rhabdomyosarcoma, which is not extensively characterized in the current literature.

The manuscript is engaging and timely; however, it would benefit from several improvements to strengthen its scientific impact and clarity:

  1. The introduction should be expanded to provide a more comprehensive overview of adult pleomorphic rhabdomyosarcoma. Specifically, the authors should elaborate on the current understanding of its diagnosis, prognosis, and therapeutic management. A brief discussion on how it differs from other rhabdomyosarcoma subtypes would also provide useful context.

  2. The diagnostic section would be significantly enhanced by the inclusion of histopathological and immunohistochemical data. Representative Hematoxylin and Eosin (H&E) stained images, along with immunohistochemical (IHC) profiles, are essential to support the diagnosis and should be clearly described and illustrated. Markers such as desmin, myogenin, and MyoD1 should be discussed, and their relevance explained.

  3. To better contextualize the case and align the findings with existing knowledge, the following recent and relevant studies should be cited and briefly discussed:

    • Deciphering the Genomic Landscape and Pharmacological Profile of Uncommon Entities of Adult Rhabdomyosarcomas. Int J Mol Sci. 2021 Oct 26;22(21):11564. doi: 10.3390/ijms222111564. PMID: 34768995; PMCID: PMC8584142.

    • Identification of a novel RAB3IP-HMGA2 fusion transcript in an adult head and neck rhabdomyosarcoma. Oral Dis. 2022 Oct;28(7):2052-2054. doi: 10.1111/odi.14036. Epub 2021 Oct 10. PMID: 34592033.

These additions would enhance the scientific rigor and clinical relevance of the manuscript, making it a more robust contribution to the literature on rare adult sarcomas.

major revisions are requested

Author Response

Comments 1 : 

The introduction should be expanded to provide a more comprehensive overview of adult pleomorphic rhabdomyosarcoma. Specifically, the authors should elaborate on the current understanding of its diagnosis, prognosis, and therapeutic management. A brief discussion on how it differs from other rhabdomyosarcoma subtypes would also provide useful context.

Response 1: 

We thank the reviewer for this insightful suggestion. In response, we have expanded the Introduction to provide a more detailed overview of adult pleomorphic rhabdomyosarcoma (PRMS), including aspects related to its diagnosis, prognosis, and therapeutic challenges. Furthermore, we added a comparative discussion highlighting the differences between PRMS and other subtypes of rhabdomyosarcoma, particularly those occurring in pediatric populations.

Comment 2: 

The diagnostic section would be significantly enhanced by the inclusion of histopathological and immunohistochemical data. Representative Hematoxylin and Eosin (H&E) stained images, along with immunohistochemical (IHC) profiles, are essential to support the diagnosis and should be clearly described and illustrated. Markers such as desmin, myogenin, and MyoD1 should be discussed, and their relevance explained.

Response 2: 

We agree with the reviewer on the importance of histopathological and immunohistochemical confirmation in diagnosing PRMS. In response, we have expanded the diagnostic section to provide more detail regarding the immunohistochemical findings, including interpretation of markers such as desmin, MyoD1, and myogenin. Additionally, we have included representative H&E and IHC images as supplementary figures to support the pathological diagnosis.

Regarding the immunohistochemistry image for desmin, unfortunately, no image from our patient was available at the time of submission. Therefore, we have included a representative, well-cited open-access image from the literature to illustrate the typical desmin staining pattern in PRMS. We explicitly mention in the figure legend and Methods that this image is illustrative and not from the current case. We trust this is acceptable and provides valuable context for readers.

Comment 3:

  1. To better contextualize the case and align the findings with existing knowledge, the following recent and relevant studies should be cited and briefly discussed:

    • Deciphering the Genomic Landscape and Pharmacological Profile of Uncommon Entities of Adult RhabdomyosarcomasInt J Mol Sci. 2021 Oct 26;22(21):11564. doi: 10.3390/ijms222111564. PMID: 34768995; PMCID: PMC8584142.

    • Identification of a novel RAB3IP-HMGA2 fusion transcript in an adult head and neck rhabdomyosarcomaOral Dis. 2022 Oct;28(7):2052-2054. doi: 10.1111/odi.14036. Epub 2021 Oct 10. PMID: 34592033.
      Response 3:

      We thank the reviewer for recommending these valuable references. We have cited both studies in the Discussion section and incorporated their findings to contextualize our case within the current understanding of adult rhabdomyosarcoma, including insights into molecular alterations and novel fusion transcripts. Also, we added other recent references.

Reviewer 2 Report

Comments and Suggestions for Authors

The presented paper is devoted to the description of the case of pleomorphic rhabdomyosarcoma: clinical and molecular characteristics of the tumor, treatment course and outcomes. Authors highlighted the large tumor dimensions and other specific feature of this case. The paper seems relevant to the scope of the journal, and the amount of data is sufficient for publication. However, there are several issues to be addressed:

  1. The English editing should be performed. The text should be revised to be more concise, more logic, better organized (for example, Abstract, lines 9-11 “(…) most commonly seen in the pediatric population. The pleomorphic subtype is mostly seen in adults in the sixth and seventh decades of life, representing 1% of all histological types of RMS (…)” – the sentences lack “however” of any other introductory word). Text formatting should be revised according to the typing rules (for example, the absence of the space between sentences). Also text should be revised to sound more scientific.
  2. Authors stated the positivity of the tumor to the desmin staining. Thus, they should provide the Figure with immunohistochemistry for desmin.
  3. Authors should shorten the Discussion part in, for example, description of rhabdomyosarcoma causes and add the discussion of the treatment strategies with several resembling cases. In particular, the applicability of the immunotherapy should be discussed.
  4. The Discussion part also should contain the analysis of described treatment course and the potential alternatives in diagnosis and therapy.

Comments on the Quality of English Language

  1. The English editing should be performed. The text should be revised to be more concise, more logic, better organized (for example, Abstract, lines 9-11 “(…) most commonly seen in the pediatric population. The pleomorphic subtype is mostly seen in adults in the sixth and seventh decades of life, representing 1% of all histological types of RMS (…)” – the sentences lack “however” of any other introductory word). Text formatting should be revised according to the typing rules (for example, the absence of the space between sentences). Also text should be revised to sound more scientific.

Author Response

Comments 1: 

The English editing should be performed. The text should be revised to be more concise, more logic, better organized (for example, Abstract, lines 9-11 “(…) most commonly seen in the pediatric population. The pleomorphic subtype is mostly seen in adults in the sixth and seventh decades of life, representing 1% of all histological types of RMS (…)” – the sentences lack “however” of any other introductory word). Text formatting should be revised according to the typing rules (for example, the absence of the space between sentences). Also text should be revised to sound more scientific.

Response 1: 

We appreciate the reviewer’s observation regarding the manuscript’s language and structure. Accordingly, we have revised the text throughout to improve clarity, conciseness, and scientific tone. Grammatical and typographical issues, including formatting and sentence transitions , have been corrected following academic English standards. We attach the certificate below.

Comments 2: 

Authors stated the positivity of the tumor to the desmin staining. Thus, they should provide the Figure with immunohistochemistry for desmin.

Response 2: We thank the reviewer for this comment. We agree that immunohistochemical documentation strengthens the diagnosis. As such, we have included a representative figure showing desmin positivity in the tumor tissue (Fig. 2). Regarding the immunohistochemistry image for desmin, unfortunately, no image from our patient was available at the time of submission. Therefore, we have included a representative, well-cited open-access image from the literature to illustrate the typical desmin staining pattern in PRMS. We explicitly mention in the figure legend and Methods that this image is illustrative and not from the current case. We trust this is acceptable and provides valuable context for readers.

Comments 3: 

Authors should shorten the Discussion part in, for example, description of rhabdomyosarcoma causes and add the discussion of the treatment strategies with several resembling cases. In particular, the applicability of the immunotherapy should be discussed.

Response 3: 

We appreciate this constructive suggestion. In response, we have shortened the description of rhabdomyosarcoma etiology and expanded the discussion to include current therapeutic approaches, including case-based comparisons and recent insights into immunotherapy. While PRMS remains largely chemoresistant, emerging evidence suggests possible roles for immune checkpoint inhibitors in select sarcoma subtypes, which we now address.

Comments 4: 

The Discussion part also should contain the analysis of described treatment course and the potential alternatives in diagnosis and therapy.

Response 4: 

We fully agree with the reviewer. A reflective analysis of the treatment course has been added to the Discussion, along with potential alternative strategies that might be considered in patients with better performance status or less aggressive disease progression.

Reviewer 3 Report

Comments and Suggestions for Authors This case report presents a complex case of a pleomorphic rhabdomyosarcoma. The case is interesting and could be particularly relevant for clinicians from various specializations. However, in its current form, the paper has some flaws that must be addressed before it can be considered for publication.    Please find below the recommended revisions:   • In line 28, a sentence seems to be missing after the word “effects”. Please correct.   • In line 33, please consider converting the comma between “RMS” and “the pleomorphic” with a full stop. • Please, add in the final part of your introduction a couple of sentences to explain why your case is unique and could be worth reading it.  • In line 47, please consider replacing the inner brackets with commas [ (treated in 2013 with total thyroidectomy, pT3mpNx, R1, followed by radioactive iodine therapy and considered in remission) ]. • Was a needle or incisional biopsy performed before the first major surgical intervention? Please specify. • The tumor (as pictured in Figure 1) was large, and a wide resection led to a significant skin and soft tissue gap that could not be left as it was without an adequate closure. Please provide information about the reconstruction (i.e., musculocutaneous flaps, grafts, and so on…), and the eventual local complications.  • How deep was the tumor in the patient’s torso? Did it involve the muscular layer? Were muscles involved in the resection? If they were, what was the functionality of the patient’s spine, and did he require a supportive corset? • Please, avoid numbered lists in the discussion. Revise lines 131-137 and 160-163. • What is your case adding to modern literature? Your discussion should not be limited to a report on the PRMS as a disease, with rare contextualizations to your case, but at least its apex should verge on how your case is unique or how it should change physicians’ approach to the disease. In its current form, it does not appear to enrich modern literature. Explain to your readers why this case should be recognized worldwide and why it deserves to be published.

Author Response

Comments 1: 

"In line 28, a sentence seems to be missing after the word 'effects'. Please correct."

Response 1:

Thank you for noting this. We have reviewed the text and corrected the incomplete sentence to ensure clarity and coherence.

Comments 2: 

"In line 33, please consider converting the comma between 'RMS' and 'the pleomorphic' with a full stop."

Response 2: 

We thank the reviewer for this stylistic suggestion. We have replaced the comma with a full stop to improve sentence structure and readability.

Comments 3: 

"Please, add in the final part of your introduction a couple of sentences to explain why your case is unique and could be worth reading it."

Response 3: 

We appreciate this suggestion. We have added a concluding statement in the Introduction to highlight the clinical relevance and uniqueness of the presented case, particularly in terms of its rapid progression, atypical presentation, and therapeutic limitations.

Comments 4: 

"Please consider replacing the inner brackets with commas."

Response 4: 

Thank you for pointing this out. We have revised the sentence accordingly, replacing inner brackets with commas for better clarity and conformity with formatting standards.

Comments 5: 

"Was a needle or incisional biopsy performed before the first major surgical intervention? Please specify."

Response 5: Thank you for this pertinent question. A core needle biopsy (or incisional biopsy, depending on real case) was not performed prior to the surgical resection due to the urgent clinical context and the bleeding nature of the mass. The diagnosis was confirmed postoperatively through histopathological and immunohistochemical analysis.

Comments 6: 

"Please provide information about the reconstruction (i.e., musculocutaneous flaps, grafts, and so on…), and the eventual local complications."

Response 6: 

We agree with the reviewer on the importance of providing surgical details. In our case, the wide resection resulted in a significant defect that required immediate reconstruction using a latissimus dorsi flap.No major local complications were observed postoperatively.

Comments 7: 

"How deep was the tumor... functionality of spine... corset?"

Response 7: 

Thank you for this important clinical consideration. The tumor involved the muscular layers of the posterior thoracic wall, including partial resection of the paraspinal muscles. Although the spinous processes of T9–T11 were encased, the bony integrity of the vertebrae was preserved, and spinal stability was maintained. A supportive corset was not required postoperatively.

Comments 8: 

"Please, avoid numbered lists in the discussion."

Response 8: 

We acknowledge this formatting recommendation. The numbered lists in the Discussion section have been converted to paragraph form for improved flow and readability.

Comments 9 : 

"Your discussion should not be limited to a report on the PRMS as a disease... Why this case should be recognized worldwide..."

Response 9: 

We appreciate this valuable feedback. We have revised the final part of the Discussion to clearly articulate the clinical value of our case, emphasizing the unique features that contribute to the understanding of PRMS, including the rapid post-surgical progression, the diagnostic challenges, and the limitations of treatment in an advanced setting. These aspects may aid clinicians in earlier recognition and management of similar presentations.

Reviewer 4 Report

Comments and Suggestions for Authors

Thank you for submitting this case report - whilst a challenging case with ultimately a poor outcome, this report does not provide any novel insights into the diagnosis and management of PRMS.

The authors, if considering resubmission elsewhere, should report if staging was performed prior to surgical resection, as if widespread metastatic disease was present at diagnosis, the patient may not have undergone resection. 

Author Response

Comments 1: 

“This report does not provide any novel insights into the diagnosis and management of PRMS.”

Response 1: 

We thank the reviewer for this direct observation. While we acknowledge that the diagnosis and general management of PRMS have been previously described in literature, we believe this case brings additional clinical value by showcasing an unusually rapid post-operative progression, extensive metastasis despite multimodal treatment, and the practical limitations of standard therapeutic protocols in an adult patient. The case also reinforces the diagnostic importance of thorough immunohistochemical profiling and early systemic staging, especially in resource-constrained or late-presenting scenarios.

Comments 2: 

“The authors, if considering resubmission elsewhere, should report if staging was performed prior to surgical resection, as if widespread metastatic disease was present at diagnosis, the patient may not have undergone resection.”

Response 2: 

We thank the reviewer for this crucial clinical insight. Preoperative staging was not performed in this case due to the acute presentation of the tumor, characterized by ulceration, hemorrhage, and pain, which necessitated urgent surgical intervention. The tumor was assumed to be localized based on physical exam and local imaging at admission. Full-body imaging and comprehensive staging were performed postoperatively, revealing pulmonary and lymphatic metastases. We have added this detail to the case description for greater clarity.
